# In Vivo Shadows and In Vitro Light: The Early Embryological Journey Amid Endometriosis

**DOI:** 10.3390/biology14080957

**Published:** 2025-07-29

**Authors:** Grzegorz Mrugacz, Aleksandra Mospinek, Maria Modrzyńska-Olejniczak, Bartłomiej Byczkowski, Ewelina Radaj, Piotr Olcha

**Affiliations:** 1Center for Reproductive Medicine Bocian, Białystok Branch, 26 Akademicka St., 15-267 Białystok, Poland; gmrugacz@klinikabocian.pl; 2Center for Reproductive Medicine Bocian, Łódź Branch, 157a Piotrkowska St., 90-440 Łódź, Poland; 3Center for Reproductive Medicine Bocian, Lublin Branch, 26 Relaksowa St., 20-819 Lublin, Polandpiotrolcha@op.pl (P.O.); 4Department of Gynecology and Gynecological Endocrinology, Medical University of Lublin, Al. Racławickie 23, 20-049 Lublin, Poland

**Keywords:** endometriosis, infertility, in vitro fertilization (IVF), oocyte quality, oxidative stress, embryo morphokinetics, transcriptomics

## Abstract

Endometriosis harms fertility by creating a hostile environment for reproduction through chronic inflammation, immune dysfunction, oxidative stress, anatomical distortion, and ovarian damage. In vitro fertilization (IVF) provides a controlled solution, albeit with limitations. This procedure bypasses the body’s hostile environment through core mechanisms such as directly retrieving eggs before they are damaged, fertilizing and growing embryos in a protected lab setting, and selecting the best embryos for transfer. IVF’s efficacy is highlighted by the fact that it significantly improves pregnancy chances compared to natural conception for endometriosis patients. Nevertheless, it still exhibits fundamental limitations despite its status as the hope for affected patients. In cases of severe disease and lower ovarian reserve, fewer eggs are retrieved and the egg/embryo quality is lower. These differences result in the IVF success rate being lower than in non-endometriosis patients. Regardless, future progress lies in personalized IVF approaches and novel strategies targeting the root causes, such as oxidative stress and inflammation, to further improve outcomes. Additionally, innovation will likely by achieved by leveraging artificial intelligence and time-lapse imaging to foster advanced embryo selection.

## 1. Background

Endometriosis is a biologically complex disorder with significant impacts on fertility, affecting approximately 10% of reproductive-aged women worldwide and 50% of female infertility cases [1]. This disorder has three major types: peritoneal, ovarian, and deep infiltrating endometriosis [1]. Each contributes to local (ovaries, fallopian tubes, and uterus) and systemic reproductive functional discrepancies through mechanisms that include anatomical distortion, immune dysregulation, chronic inflammation, and oxidative stress [1,2,3,4,5,6,7,8,9,10,11,12,13,14,15]. Oxidative stress, as a standalone mechanism, underlies cellular damage [16,17,18,19,20]. The paradox of endometriosis is its variable impact on fertility. Some women can get pregnant naturally, while others struggle and need assisted reproductive technologies (ARTs), such as in vitro fertilization (IVF) [21,22].

Emerging evidence highlights the detrimental effects of endometriosis on oocyte quality, ovarian follicle maturation and development, and early embryo development [22,23,24]. These effects are driven by oxidative stress and altered transcriptomic profiles [24,25,26,27,28,29]. However, IVF offers a controlled environment to bypass these in vivo obstacles [30]. While it does so, debates continue regarding its efficacy due to variable outcomes related to fertilization rates, embryo quality, and aneuploidy risks [5,6,31,32,33,34,35,36,37,38]. This article explores the interplay between endometriosis’s in vivo pathophysiology and the in vitro rescue potential of ART, synthesizing current evidence to guide clinical practice and future research.

## 2. The In Vivo Impact of Endometriosis on Fertility

Endometriosis disrupts fertility through multiple mechanisms, including alteration of normal anatomical structures, hormonal fluctuations, and immune dysfunction. Below is an overview of the pathways.

### 2.1. Pathophysiology and Classification

Endometriosis is classified into three main types, and each contributes differently to infertility. The Revised American Society for Reproductive Medicine Classification of Endometriosis (rASRM) staging system and the Endometriosis Fertility Index are leveraged for the classification of the three types, which helps predict reproductive outcomes, while the Enzian classification focuses on deep lesions [2,7,9,13].

Peritoneal endometriosis alters pelvic biochemistry [16,19,39,40,41]. Based on the relevant literature, it is the most common and least severe type of endometriosis. In this type, small lesions form on the peritoneum, as seen in Figure 1 below, and pelvic biochemistry is altered through pro-inflammatory cytokines and prostaglandins. This inflammation, in turn, impacts egg viability and implantation. Oxidative stress also plays a major role, as reactive oxygen species (ROS) damage oocytes, sperm, and embryos [16,25,39,40,42,43,44]. The most common mitigation strategy for this type of endometriosis is medical therapy. Based on the available evidence, 3–6 months of administering gonadotropin-releasing hormone (GnRH) agonists is expected to suppress inflammatory cytokines and prostaglandins [5,6]. Further, antioxidants such as melatonin and N-acetylcysteine are employed to alleviate oxidative stress.

Ovarian endometriosis (endometriomas) damages the ovarian stroma, which, in turn, reduces the follicular reserve [17,18]. Chocolate cysts (Figure 2) form on the ovaries, damaging ovarian tissue and, in the process, limiting the healthy follicles available for ovulation. Iron toxicity is another evident mechanism of this type of endometriosis [41]. Basically, endometrioma fluid contains hemosiderin, which induces oxidative apoptosis in granulosa cells [41]. Lastly, the presence of cysts may necessitate cystectomy, a process that further reduces the antral follicle count (AFC), thereby negatively impacting fertility [35,46,47]. Mitigation of this endometriosis type includes cystectomy, which in itself further hinders fertility [17,35,48]. Sclerotherapy may also be used to minimize follicular loss [47].

Deep infiltrating endometriosis (DIE) impacts uterosacral ligaments and the bowel, causing structural alteration [15,20,21]. This is the most severe form of endometriosis. It proliferates approximately 5 mm into the surrounding tissues, hence the structural alteration [20,21]. With the alterations, scarring and anatomical distortions are likely to be evident, as seen in Figure 3 below, minimizing the chances of conception [50,51,52]. Another mechanism evident in DIE is neuroangiogenic invasion. This entails lesions infiltrating nerves and blood vessels, thereby increasing the chances of dyspareunia and dyschezia [20,21]. DIE treatments include laparoscopic excision, which carries a high risk of bowel or bladder injuries [20,21]. Preoperative imaging can also be leveraged to map lesions [9,14]^.^

In all three types, there are several proven medical therapies that mitigate the negative outcomes and increase fertility chances. However, their efficacy is limited. For instance, in the case of ovarian endometriosis, cystectomy is recommended. However, it carries a risk of ovarian damage, thereby limiting fertility chances, a core issue that the procedure is intended to solve. Surgery is also leveraged for DIE, and similar to the surgery for ovarian endometriosis, it also increases injury risks that are detrimental to conception. Hence, a notable pattern for the two serious types of endometriosis is a lack of significant and effective in vivo mitigation therapies. Could in vitro treatments be the answer? After all, ART, specifically IVF, is recommended as a significant strategy [5,26,35,43,54,55,56,57,58]. A further look into in vitro treatment alternatives is therefore warranted to evaluate the hope they hold for women affected by endometriosis.

NB: Staging systems 1–14 guide prognosis but fail to fully capture functional impacts like oocyte quality or the embryotoxicity caused by oxidative stress.

### 2.2. Reproductive Consequences

Endometriosis impairs fertility via the following mechanisms:

Ovarian reserve: Fibrosis and oxidative damage caused by endometriomas underlie reductions in the antral follicle count (AFC) and the levels of anti-Müllerian hormone (AMH) [17,18,24,25,47,59,60,61,62,63,64,65,66,67,68,69,70,71]. In fibrotic destruction, endometriomas replace a healthy ovarian stroma with fibrous tissue. On the other hand, oxidative stress triggers apoptosis in granulosa cells and primordial follicles [59,60,61,62,63,72]. This results in declines in the indicated biomarkers, reflective of diminished ovarian reserve [47,69,70]. The clinical implication of this is early IVF consideration due to the higher risk associated with cystectomy [55,56,57,58].

Folliculogenesis: Dysregulated steroidogenesis and altered follicular microenvironments, such as iron overload in endometriomas, compromise oocyte maturation [17,18,23,24,47,64]. Specifically, the iron-dense microenvironment damages granulosa cell mitochondria and disrupts oocyte meiosis [24,64,71,73]. The clinical implication of this mechanism is adjuvant therapy consideration. As highlighted in the relevant literature, antioxidants such as melatonin are needed to mitigate oxidative stress in vivo and in vitro [58,59,60,61,62,63,72,74,75].

Anatomical disruption: Adhesions distort tubo-ovarian anatomy, impairing ovulation and embryo transport [16,17,18,20]. Specifically, with adhesion formation, inflammatory cytokines underlie fibrosis. This, in turn, blocks embryo transport and prevents ovum pick up [16,17,20]. Clinically, natural conception can be improved in mild cases [16,44]. However, IVF is necessary in cases of severe tubal distortion [5,6].

Inhibited sperm motility: Peritoneal fluid cytokines impact sperm motility [16,19].

DNA damage: In sperm, oocyte, and endometrial cells specifically, besides the DNA, lipids and proteins are also damaged. ROS impact the cell membrane [76,77,78,79], disrupting spindle formation. The resultant DNA fragmentation affects embryonic development potential [80,81,82,83]. Further, endometriosis alters RNA expression profiles in oocytes [81]. This results in dysregulated gene expression underlying lower development chances [8,54]. Morphological abnormalities, such as cytoplasmic granularity, zona pellucida thickness, polar body fragmentation, and an enlarged perivitelline space, as seen in Figure 4 below, are more frequent [30,31,33,44,84], potentially affecting fertilization and early embryo progression. The clinical implication of this is IVF consideration, which is necessary because it allows sperm selection and embryonic protection using a medium that is antioxidant-rich [85].

Clinical paradox: Even mild endometriosis (stages I–II) can cause infertility without obvious anatomical defects, implicating non-mechanical factors [2,3,6,7,8,9,10,11,12,13]. However, in cases where the impact is not severe, there is hope for natural conception after corrective surgical measures. In cases where the damage is severe, IVF remains a significant recommendation, prompting a further look into its viability.

### 2.3. Immune Dysregulation and Inflammation

Chronic inflammation in endometriosis drives infertility through the following mechanisms:

Peritoneal fluid alterations: The volume of peritoneal fluid is significantly higher in infertile women with endometriosis than in those without this disease [87]. Increased macrophages, cytokines (Interleukin-6 (IL-6) and Tumor Necrosis Factor-α (TNF-α)), and prostaglandins create unconducive conditions for gametes and embryos [16,60].

Autoimmune components: Antibodies against endometrial antigens may disrupt implantation [22,87,88,89,90]. Specifically, the humoral immune response is enhanced, with increases in B-lymphocyte numbers and autoantibody production [87,88,89,90]. Based on immunohistochemical and gene expression microarray analysis, it was shown that endometriotic lesions were abundant in plasma cells and activated macrophages with the highly expressed cytokine B-lymphocyte stimulator [88,90].

Progesterone resistance and increased estrogen: Decidualization is impaired due to inflammatory signaling [89,91]. Estrogen stimulates cyclooxygenase-2 enzyme (COX-2) expression, which leads to prostaglandin E2 (PGE2) synthesis. Production of PGE2 and cytokines facilitates infertility in women with endometriosis [89].

### 2.4. Cellular Actors and Pathways

Macrophage polarization: This occurs through M1 dominance in peritoneal fluid and is specifically caused by TNF-α and IL-6 [16,60]. Particularly, pro-inflammatory M1 macrophages (CD68^+^/CD80^+^) outnumber anti-inflammatory M2 macrophages by a ratio of 3 to 1 [16,60]. TNF-α increases FS7-associated cell surface antigen (FAS) receptor expression on granulosa cells, resulting in apoptosis via the FAS/FAS Ligand pathway, a crucial mechanism for programmed cell death within the immune system [22]. This disrupts the gap junction connexin-43 in cumulus–oocyte complexes [73,83]. On the other hand, IL-6 activates STAT3 (Signal Transducer and Activator of Transcription 3), which, in turn, causes premature luteinization of granulosa cells. The result is reduced aromatase activity, which causes an abnormal estrogen/progesterone ratio [23].

Complement system activation: C3a/C5a deposition on the zona pellucida results in impaired sperm binding [89]. Basically, C3a/C5a binds ZP2/ZP3 glycoproteins. This causes structural masking of sperm receptors, thus triggering premature cortical granule exocytosis. There is also the membrane attack complex (MAC), where CD59 deficiency on the oolemma causes C5b-9 pore formation. This cascades into a calcium overload that underlies oocyte activation failure.

Natural Killer (NK)-cell dysfunction: In this mechanism, perforin/granzyme B in follicular fluid underlies oocyte DNA fragmentation [91]. This process is labeled as cytotoxic overactivation, where perforin/granzyme B levels are 2 to 3 times higher in follicular fluid [91]. The outcome is direct oocyte DNA cleavage via caspase-3 activation. Further, lamin B1 is targeted, resulting in nuclear envelope breakdown. In addition, there is the uterine natural killer (uNK) paradox, where peripheral NK cytotoxicity increases while endometrial uNK function proportionally decreases. The result is impaired angiogenesis at the implantation site and reduced trophoblast invasion support.

The impact of the above actors and pathways on the oocyte and embryo is cumulus cell inflammation, which underlies disrupted gap junctions [73]. Further, TLR4 (Toll-like receptor 4)/NF-κB (nuclear factor kappa B) activation in oocytes fosters premature meiotic resumption [25].

Evidence: Women with endometriosis show higher levels of inflammatory markers in their follicular fluid. This, as evidenced, correlates with poor oocyte quality [22,40,59,82,89,90,91].

### 2.5. Oxidative Stress and Free Radicals

Reactive oxygen species (ROS) are central to endometriosis-related infertility:

The follicular microenvironment: Iron from endometriomas generates ROS, which, in turn, damage oocyte mitochondria and spindle integrity [24,39,41,58,64,89].

Cumulus–oocyte complex (COC) dysfunction: ROS impair gap junctions. This underlies both structural and functional abnormalities that impede oocyte maturation, fertilization, and the subsequent development of the embryo [22,25,73,78,79,91].

Embryotoxicity: Oxidative stress induces DNA fragmentation, lipid and protein damage, and apoptosis in early embryos [58,61,62,66,67,68,74,78,81,89].

Therapeutic implications: Antioxidants such as vitamin E show promise in animal models but lack robust clinical data [67,71,72].

### 2.6. Cellular Damage: Mechanisms of ROS Damage

Mitochondrial dysfunction: The first significant mechanism is mtDNA deletions. They occur in oocytes and are fostered by iron-catalyzed Fenton reactions [24,71]. For these reactions, endometrioma fluid releases free iron (Fe^2+^), which reacts with hydrogen peroxide (H_2_O_2_) to generate hydroxyl radicals (OH). The hydroxyl radicals are a powerful oxidizing agent, underlying ROS’ impact on oocyte quality [74]. Another significant mechanism is impaired oxidative phosphorylation, which refers to reduced ATP production for meiosis [25,71].

Lipid peroxidation: In this mechanism, ROS attack polyunsaturated fatty acids in oocyte membranes. Specifically, powerful hydroxyl radicals attack omega-3 polyunsaturated fatty acids (PUFAs) in the oolemma, forming lipid peroxides such as malondialdehyde that, in turn, disrupt membrane fluidity. The result is impaired calcium oscillation patterns and sperm–oocyte fusion [25,89]. This mechanism also compromises spindle assembly since peroxidized tubulin polymers cause abnormal microtubule assembly [24]. As such, aneuploidy risk is high, even in morphologically normal oocytes.

Epigenetic alterations: ROS-induced 8-oxoguanine lesions in DNA underlie abnormal methylation in zygotes [83]. Alternatively, epigenetic alterations also act through histone modification defects via redox-sensitive enzymes such as sirtuins [54]. Particularly, deacetylation of H4K16 underlies chromatin compaction defects [54]. This disrupts transcriptional silencing in zygotes. Below is Table 1 providing a summative review of key pathways affected by epigenetic alterations.

Overall, the key pathways affected by these three ROS damage mechanisms are NRF2 (nuclear factor erythroid 2-related factor 2)/KEAP1 (Kelch-like ECH-associated protein 1) antioxidant system suppression [62], HIF-1α (Hypoxia-inducible factor 1-alpha) stabilization, and a pseudohypoxic follicular microenvironment [24].

### 2.7. In Vivo Summary

In vivo, endometriosis disrupts fertility through anatomical, inflammatory, and oxidative pathways. Several treatment strategies that try to address the fertility dilemma in situ are recommended. For instance, GnRH agonists and antioxidants are recommended for peritoneal endometriosis. For ovarian endometriosis and DIE, surgical procedures are the recommended strategies. All have varying degrees of success, thriving mostly in milder cases. However, in situ treatment strategies are significantly limited. Worse, they exhibit severe side effects, such as ovarian cancer risk, that potentially make fertility unattainable. To counter their negative outcomes and improve fertility chances for affected women, in vitro treatment is highly recommended [5,26,35,43]. The next section of this article evaluates the in vitro alternative, scrutinizing any fertility hopes it carries for affected women.

## 3. The In Vitro Alternative: Controlled Rescue

Endometriosis creates a hostile reproductive environment through various mechanisms that IVF can significantly alleviate [43,54]. One way is through bypassing anatomical barriers, as IVF overcomes tubal dysfunction and adhesive disease. These two conditions account for 30–50% of all endometriosis cases, and IVF overcomes them by avoiding the need for natural gamete transport [16,20,91]. This avoidance eliminates gamete exposure to the toxic peritoneal fluid microenvironment [24,25,60]. Further, IVF fosters ovarian stimulation advantages. Specifically, controlled hyperstimulation helps free follicles from the oxidative endometriotic environment earlier in their development [47,69]. One mechanism through which IVF acts is liberating the oocyte.

### 3.1. Liberating the Oocyte

Inflammation and oxidative stress impair natural conception [5,6]. IVF avoids this outcome by bypassing the endometriosis-related hostile peritoneal environment. The procedure entails retrieving oocytes directly from follicles, either mechanically (Figure 5) or through enzymatic digestion. As such, IVF serves as both a diagnostic tool (revealing oocyte quality deficits) and a therapeutic intervention, particularly for patients with compromised tubal function or severe disease [26,27].

Physically, the oocyte is removed from the iron-dense follicular fluid in endometriomas, a pro-inflammatory peritoneal cavity, and hypoxic conditions near deep lesions [16,17,20,21,60,64]. By bypassing structural barriers, IVF fosters avoidance of adhesion-distorted tubo-ovarian anatomy, endometrioma-related mechanical obstruction, and altered tubal motility [16,17,20,47,91]. Several technical innovations enhance oocyte liberation through IVF, including ultrasound-guided aspiration (Figure 5), which, as evidence shows, has a 23% higher oocyte yield with 3D ultrasound guidance [47]. Others include cumulus cell protection, which has advantages such as meiotic regulation, and follicular flashing optimization, which recovers 18% more oocytes in endometriosis and reduces oxidative damage during retrieval [24,35,73].

The biological consequences of liberation include metabolic programming. That is, the in vitro platform fosters escape from hypoxic follicular conditions and normalization of nutrient gradients [58,64,80]. Another consequence is epigenetic rescue, where irregular DNA methylation and histone modification defects are prevented [28,54,83].

The efficacy of this process is highlighted by significant clinical evidence. For instance, compared to in situ, there is a 68% reduction in the oocyte apoptosis rate post-retrieval [24]. Further, there is an over 2-fold improvement in maturation competence, and restoration of normal spindle morphology has been recorded in 73% of cases [24,30,33,48]. However, despite its success, oocyte liberation faces significant persistent challenges that have plagued its hope-providing capacity among affected women, where in situ treatment is no longer an alternative. One of the key operational challenges is residual cellular memory. Based on the available evidence, transcriptomic alterations are likely to persist despite retrieval [28,54]. Further, mitochondrial DNA damage is also likely to remain [25,71]. Another significant challenge is technical limitations. For instance, existing ovarian reserve depletion cannot be reversed [47,69]. Alternatively, there is a high risk of endometrioma rupture during retrieval [35].

Overall, oocyte liberation improves gamete viability through escape from the damaging microenvironments of endometriosis. However, the much-leveraged IVF alternatives still exhibit significant limitations in their scope, necessitating integration with advanced lab techniques for complete biological rescue.

### 3.2. IVF Efficacy

IVF exhibits efficacy by fostering oocyte retrieval benefits and embryo culture advantages. The oocyte retrieval benefits, as noted earlier, include preventing exposure to iron-rich endometrioma fluid. Oocyte retrieval shows fertilization rates that are 2 to 3 times higher compared to natural conception attempts, as per studies [35,37]. For the embryo culture advantages, lab conditions ensure culture media free of ROS, optimal hormone levels that tend to be absent as a result of endometrial dysregulation, and selection of the best embryos through time-lapse monitoring [23,24,32,48,58,73]. All these attributes underlie IVF’s observed efficacy, as demonstrated by the following markers: Cumulative live birth rates after three cycles are 40–55%, compared to 60–65% in tubal factor infertility. Success per cycle is 22–28% for mild/moderate conditions and 15–20% for severe cases [35,46,57]. For comparable success, 1.8 times as many cycles are required compared to non-endometriosis patients [55].

For stage-dependent outcomes, the live birth rate is 25–30%, which makes it comparable to tubal factor infertility in minimal/mild cases [23,56]. For moderate cases, there is a 15–25% success reduction compared to controls [35,37]. For severe cases where pretreatment is not considered, the live birth rate is the lowest at, 12–18% [20,57]. When it comes to ovarian response patterns, 20–30% fewer oocytes are retrieved in endometrioma cases [47], and they have a 15% lower maturation rate [30,33]. The fertilization rates are 65–70%, compared to 75–80% in controls [37,50]. For embryo development metrics, IVF exhibits 30% slower early divisions and two times as much fragmentation in terms of cleavage anomalies [33,48]. The blastocyst formation rate is 35–40%, compared to 50–55% in controls [37,50]. However, achieving quality blastocysts is still manageable [57]. With respect to other factors, cystectomy leveraged for ovarian endometriosis reduces success by 40% [35] and a 6-month GnRH agonist pretreatment regimen, which is mostly used for peritoneal endometriosis, improves the live birth rate by 3.2-fold [6,57]. Further, IVF has 3.1-fold better outcomes if it follows laparoscopy [6,35].

Persistent challenges are also evident, even with optimized protocols. For instance, the maximal live birth rates remain 10–15% lower when compared to non-endometriosis patients [35,57]. Further, aneuploidy rates remain similar, but the implantation potential is reduced [56,57]. Overall, IVF significantly improves fertility outcomes in endometriosis. However, its efficacy remains limited by disease severity and the ovarian reserve status. Evidence-based protocol modifications can optimize success. Nevertheless, biological limitations highlight the need for continued innovation in ART approaches for endometriosis patients. Table 2 below summarizes the treatment’s efficacy.

For better outcomes, based on the above efficacy evaluation, IVF should be started early, severe cases should be pretreated with GnRH agonists or laparoscopy, aggressive stimulation should be considered for expected poor responders, and time-lapse embryo selection should be considered as a compensatory measure for morphokinetic alterations.

### 3.3. Conflicting Evidence: The IVF Outcome Debate

Oocyte/embryo quality impairment: Meta-analyses and cohort studies associate endometriosis, particularly endometriomas (OR: 0.62, 95% CI: 0.48–0.79), with reduced fertilization rates (15–20%), poorer embryo quality, and lower pregnancy success [33,35,36,37,50].

Embryo development: IVF has been shown to have 30% slower cleavage rates and 40% lower blastocyst formation [37,48,50].

Pregnancy outcomes: A 25% reduction in the live birth rate is observed in cases of severe endometriosis [35,57].

Neutral/no effect: Some studies report comparable IVF outcomes in endometriosis patients, with no increased aneuploidy rates [42,43,56,57]. Further, the implantation potential is comparable when transferring preimplantation genetic testing for aneuploidy (PGT-A)-screened embryos [57].

Aneuploidy: No conclusive evidence links endometriosis to higher embryonic aneuploidy. This suggests that other mechanisms, such as oxidative stress, may dominate [56].

Oocyte donation studies: The live birth rate is similar when using donor oocytes [42,43].

Lab factors: Outcomes improve with antioxidant-supplemented media [58,62]. However, standard media may exaggerate deficits [25].

Cohort studies and meta-analyses: Heterogeneity in study designs and patient populations highlights the need for standardized staging and larger datasets [34,46,55].

### 3.4. IVF Conclusion

While IVF mitigates some endometriosis-related barriers, oocyte and embryo quality challenges remain evident. Outcomes vary by disease severity and individual response. Hence, further research is needed to reconcile conflicting evidence and optimize ART protocols. The conflicting evidence highlights the biological complexity of endometriosis. While a research consensus confirms reduced efficacy in severe cases, more so with endometriomas, optimal ART strategies can mitigate many disadvantages. Hence, further research should focus more on personalized protocols.

Overall, the above analysis confirms that, in general, IVF has significant limitations that impact its efficacy in terms of improving fertility and live birth rates. However, its arsenal includes time-lapse technology, a rather advanced tool aiding the detection of fundamental markers associated with embryo viability and implantation. The next section of this article evaluates how embryo morphokinetics can be leveraged as a supplement to standard morphological assessment to improve IVF outcomes.

## 4. Embryo Morphokinetics: Time-Lapse and Timing in Endometriosis

### 4.1. Introduction to Time-Lapse Imaging in Embryo Selection

Time-lapse microscopy (TLM) has revolutionized in vitro embryo assessment by enabling continuous, non-invasive monitoring of developmental dynamics. Compared to traditional static evaluations, TLM captures critical morphokinetic parameters. These include cleavage timing, blastomere symmetry, and blastocyst formation. The outcome is a more nuanced understanding of embryo viability [32,48]. The 2010 Istanbul Consensus on Oocyte Cryopreservation established standardized criteria for embryo assessment [32]. However, TLM further refines selection by identifying subtle temporal patterns that predict an increased chance of implantation.

### 4.2. Morphokinetic Profiles in Endometriosis: Altered Dynamics?

Emerging evidence suggests that endometriosis may impact early embryogenesis, as reflected in delayed or dyssynchronous morphokinetics:

Cleavage delays: Embryos from endometriosis patients often exhibit a prolonged time to the first cleavage and irregular blastomere division intervals. These differences may be due to oxidative stress or mitochondrial dysfunction [48,51].

Blastocyst formation: Some studies report slower compaction and blastulation rates. However, blastocyst quality may remain comparable to controls [37,48].

Conflicting data: A sibling oocyte study found no significant morphokinetic differences in embryos derived from endometrioma-affected ovaries [48]. This suggests that oocyte competence may not always be compromised. Conversely, meta-analyses highlight trends toward impaired development, particularly in severe endometriosis [37,51].

### 4.3. Clinical Implications and Unanswered Questions

Prognostic value: TLM may help identify endometriosis-specific morphokinetic signatures that correlate with reduced implantation.

Limitations: Heterogeneity in endometriosis subtypes, such as peritoneal and deep infiltrating endometriosis, and small sample sizes necessitate larger, stratified studies.

Future directions: Integrating TLM with oxidative stress biomarkers or transcriptomic data is recommended. This could clarify whether observed delays reflect reversible stress or irreversible oocyte damage [28,54].

### 4.4. TLM Summary

TLM reveals subtle but potentially significant alterations in embryo morphokinetics among endometriosis patients. However, its clinical relevance remains debatable. As such, standardized protocols and targeted research are needed to determine whether these deviations predict IVF outcomes. If they do, what is the right way to intervene? For now, TLM remains a fundamental research tool to identify the impacts of endometriosis in the early stages of embryogenesis.

The overall usefulness of IVF still needs further scrutiny, specifically how it can be advanced to foster personalization.

## 5. Oocyte Donation Models: Disentangling Nature from Nurture

Oocyte donation studies provide unique insights into whether endometriosis-related infertility stems primarily from oocyte quality (genetic/epigenetic alterations, mitochondrial dysfunction, and oxidative damage accumulation—Nature) or the uterine environment (endometrial receptivity defects, immune dysregulation, and anatomical distortion—Nurture).

Oocyte donation vs. own-oocyte IVF: Studies comparing IVF outcomes using donated oocytes from healthy women versus autologous oocytes in endometriosis patients reveal critical findings:

Similar pregnancy rates are observed when endometriosis patients receive donor oocytes [42,43]. This suggests that the uterine environment may remain receptive despite the disease.

Reduced success with patients’ oocytes implies innate oocyte deficits, most likely due to oxidative damage and transcriptomic errors, are a key limiting factor [28,54].

Uterine vs. cytoplasmic contributions:

Uterine factor: Endometriosis can cause inflammation and adhesions. However, the uterus often retains implantation capacity if viable embryos are transferred [43,57].

Cytoplasmic/oocyte factor: Poorer embryo development with patient-derived oocytes is likely due to functional abnormalities in mitochondria, an imbalance between ROS and antioxidant defenses, and epigenetic dysregulation [25,58,62,71,76,77].

Conclusion: Oocyte donation models highlight that oocyte quality is the predominant challenge in endometriosis-related infertility when compared to uterine receptivity. This necessitates prioritizing oocyte rescue strategies over uterine-focused interventions. Alternatively, it validates IVF as the optimal treatment that bypasses oocyte damage. Hence, amidst IVF failure, the focus should not be on the uterus as the problem but rather on how the team can improve or bypass oocyte quality limitations. Regardless, further advancement of the treatment mechanism is still needed to bypass most of its current limitations, which were highlighted earlier in the IVF Efficacy Section.

## 6. Limitations, Gaps, and Future Horizons in Endometriosis-Related Infertility

### 6.1. What We Do Not Yet Know

A lack of longitudinal IVF data: Current evidence has failed to stratify outcomes by endometriosis stage, subtype, or prior treatment. This makes it difficult to predict ART success for individual patients.

An incomplete understanding of mechanisms: Oxidative stress and inflammatory signaling are implicated [68,74]. However, their precise impacts on oocyte aging, mitochondrial dysfunction, and epigenetic changes remain unclear.

### 6.2. Emerging Diagnostic and Therapeutic Innovations

AI in embryo assessment: Artificial intelligence (AI) is presently the big thing. Machine learning algorithms may improve embryo selection by analyzing morphokinetic patterns and predicting viability beyond human grading.

Targeted therapies:

Oxidative stress mitigation: Endogenous antioxidants, such as glutathione; exogenous antioxidants, such as coenzyme Q10, melatonin, and vitamin E; culture conditions, such as mineral oil overlay; and experimental strategies, such as exosome therapy, could protect oocytes/embryos [58,59,60,61,62,63,85].

Immunomodulation: Stem cell therapy may restore endometrial receptivity. This treatment has potential due to its immunomodulatory effects and tropism toward inflamed lesion foci [87,93,94].

Receptivity biomarkers: ERA (Endometrial Receptivity Array) and BCL6 (B-cell lymphoma 6) testing could personalize embryo transfer timing.

### 6.3. Omics and Molecular Breakthroughs

Single-cell technologies: Transcriptomics and proteomics reveal dysregulated pathways in endometriosis-affected oocytes [28,54]. This paves the way for precision interventions.

Multi-omics integration: Combining genomic, metabolic, and epigenetic data may uncover new and advanced therapeutic targets and biomarkers.

Future outlook: Bridging these gaps requires collaborative, broad studies with standardized protocols. Further, innovations in AI, omics, and targeted therapies are promising for transforming care in endometriosis-associated infertility.

## 7. Conclusions: Navigating Endometriosis-Related Infertility

Endometriosis remains significant in reproductive medicine. Its systemic and local effects, which include chronic inflammation, oxidative stress, and anatomical disruption, amalgamate to impair fertility. Current evidence indicates that IVF offers a powerful workaround. However, intrinsic oocyte deficits and inconsistent outcomes highlight the complexity of this disease.

Key takeaways for clinical practice: Personalized ART strategies: Tailoring protocols based on the endometriosis subtype and severity may optimize outcomes.Oocyte quality as a priority: Emerging data emphasize cytoplasmic competence over uterine factors. This necessitates focusing on oxidative stress mitigation and advanced embryo selection tools, such as leveraging AI. These strategies would facilitate better IVF outcomes than what is currently in place.Patient-centered care: Patients need transparent counseling about the variability in IVF success rates and related innovative options. This is critical to managing expectations.

### The Road Ahead

Advances in medical science are steadily shining new light on endometriosis using avenues such as single-cell omics, AI-driven embryology, and targeted therapies. They are paving the way for breakthroughs. Nevertheless, gaps are still evident, especially in longitudinal data and standardized staging. To address them, collaborative research efforts are necessary. With respect to treatment, IVF is still reputable. However, further innovations to foster better personalization are still needed. As things currently stand, IVF should be started early; severe cases should be pretreated, as recommended in the literature; aggressive stimulation should be considered for expected poor responders; and time-lapse embryo selection should be considered as a compensatory measure for morphokinetic alterations.

For patients, it is about more than the medical challenges: it is about hope, perseverance, and resilience. Evolution in science brings with it the promise of an endometriosis-free reproductive future. Guided by compassion and innovation, the focus is to transform this promise into reality, which this article helps achieve.

## Figures and Tables

**Figure 1 biology-14-00957-f001:**
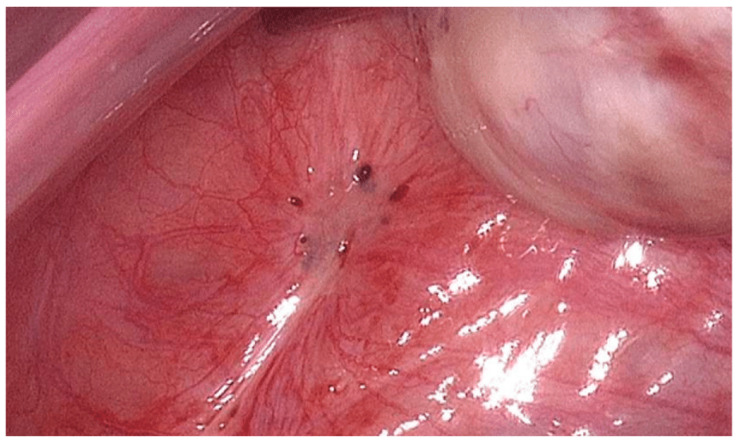
This figure was adapted from Schippert et al., *BMC Women’s Health*, 2020, under CC BY 4.0. Peritoneal endometriosis is characterized by distinct bluish lesions and an increased vascular pattern [45].

**Figure 2 biology-14-00957-f002:**
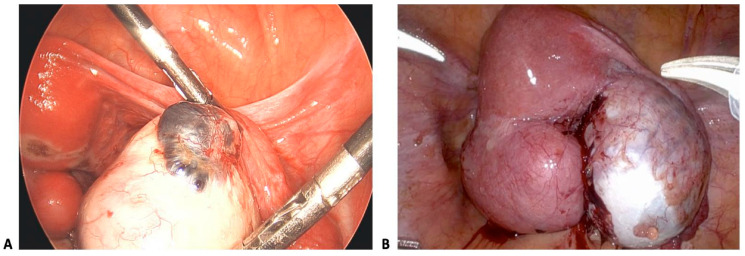
This figure was adapted from Seraji et al. (2023) under CC BY 4.0. (**A**) shows a type I endometrioma characterized by a cyst size of less than 3 cm. It also has an adherent cyst capsule, which arises from deep infiltration of superficial endometriosis implants [49]. (**B**) shows a type II endometrioma characterized by a cyst size of more than 3 cm [49]. It contains gelatinous blood clots and an easily delineated cyst wall. This endometrioma arose from physiologic functional cysts of the ovary that were invaded by endometriosis [49].

**Figure 3 biology-14-00957-f003:**
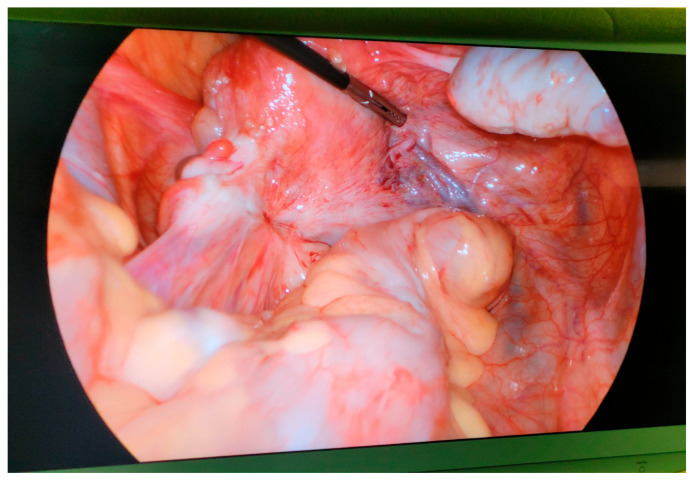
Laparoscopic view showing deep infiltrating endometriosis (DIE) involving the left ureter and surrounding pelvic structures [53]. Reproduced from Jozwik et al. (2024), under CC BY 4.0 [53].

**Figure 4 biology-14-00957-f004:**
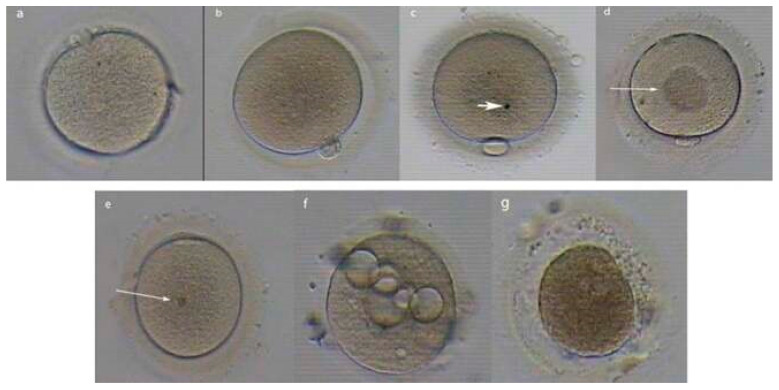
Adapted from: Nazari S. et al., under CC BY 3.0 [86] (**a**) Fragmented polar body. (**b**) Wide perivitelline space (PVS). (**c**) Refractile body—indicated by the arrow. (**d**) Central granularity—indicated by the arrow. (**e**) Bull’s eye—indicated by the arrow. (**f**) Several large vacuoles. (**g**) Degenerated cell. These abnormalities were observed by light microscopy [86].

**Figure 5 biology-14-00957-f005:**
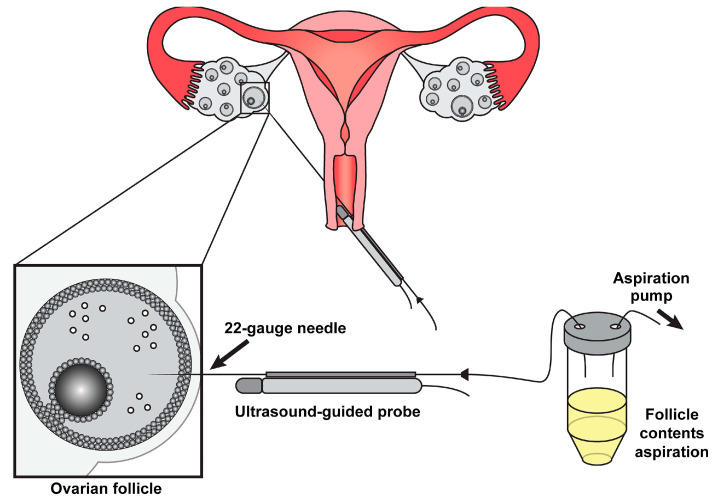
A graphical representation of mechanical oocyte retrieval adapted from Wyse et al., (2023), licensed under CC BY 4.0 [92]. Specifically, this is the ultrasound-guided aspiration technique [92].

**Table 1 biology-14-00957-t001:** Impacts of reactive oxygen species (ROS) on key developmental processes: molecular ROS targets and associated downstream effects.

Process	ROS Target	Downstream Effect
Meiosis	SOD2 (Superoxide Dismutase 2)	Superoxide increases in metaphase II oocytes, impacting oocyte quality and fostering pertinent aging
Embryogenesis	PRDX (peroxiredoxin) ½	Loss of redox homeostasis
Imprinting	DNMT3A (DNA methyltransferase 3 alpha)	Abnormal methylation patterns

**Table 2 biology-14-00957-t002:** Comparison of IVF outcomes, treatment protocols, and associated costs in patients with mild–moderate and severe endometriosis versus non-endometriosis controls.

Parameter	Mild–Moderate Endometriosis	Severe Endometriosis (DIE/Endometriomas)	Non-Endometriosis Controls
Live Birth Rate Per Cycle	22–28%	12–18%	30–35%
Oocytes Retrieved	Slightly reduced (8–12)	Significantly reduced (4–8)	10–14
Fertilization Rate	70%	60–65%	75–80%
Blastocyst Formation	35–45%	25–35%	50–55%
Euploid Embryo Rate	Comparable to controls	Comparable to controls	50–60% (age-matched)
Implantation Rate	20–25%	12–18%	25–30%
Optimal Protocol	Antagonist + standard FSH	Long GnRH agonist + high FSH	Standard protocols
Adjuvant Therapies	Optimal antioxidants	GnRH agonists (3–6 months) + antioxidants	Rarely needed
Cost Per Live Birth	USD 35,000	USD 45,000	USD 28,000

## Data Availability

Not applicable.

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
