# Peer review of "In Vivo Shadows and In Vitro Light: The Early Embryological Journey Amid Endometriosis"

_biology, 2025, doi:10.3390/biology14080957_

Round 1

Reviewer 1 Report

Comments and Suggestions for Authors

Dear Authors,

I read your review with great interest. I firmly believe that not enough attention is being given to inferlity in general especially in the context of EMS. In general medicine is way behind with female centered research, but with respect to infertility, I find that many medical systems and goverment policies are quite hipocritical as on one side they complain that birth rates are falling but on the other hand do not finance IVF or reseach enough to compensate the issue. Nevertheless, I made my comments directly on the pdf file. 

I would also like to mention that the manuscript would greatly benefit from some graphical depictions of the main concepts and issues discussed, so to highlight the most important issues. Maybe one graphical abstact or a few figures for each main chapter would be welcomed.

Author Response

Comments 1: [always in italic]

Response 1: Thank you for your comment. We have carefully reviewed the manuscript and ensured that all instances of in vivo and in vitro are now consistently italicized throughout the text, in accordance with standard scientific writing conventions.

Comments 2: [this doesn't sound quite right. maybe you could rephrase to increase readability?]

Response 2: Thank you for pointing this out. We have rephrased and broken the statement into two to increase readability 

Comments 3: [I would also mention the risk of developing ovarian cancer since it is being increasingly acknowledged that the risks are higher than previously anticipated in the context of the lack of biomarkers, or screening guidelines and treatment
guidelines that balance fertility sparring and EMS treatment (similar to those of BRCA germline mutations) 10.1001/jama.2024.9210 10.1016/j.tranon.2025.102367 Moreover, stimulation therapy for IVF is also accompanied by an increase in cancer in certain studies and conditions https://academic.oup.com/humupd/articleabstract/19/2/105/582989?redirectedFrom=fulltext&login=false. I don't believe there are any studies including women with EMS in these statistics and if their risks are different. This is a line of research with more questions than answers]

Response 3: Thank you for this suggestion. However, we just mentioned the risk of ovarian cancer without digging deeper into the pertinent mechanisms. This is because the segment is a recap of EMS on infertility and the suggestion had not been adequately explored before

Comments 4: [table title goes above the table]

Response 4: Thank you for pointing this out. We have revised the manuscript accordingly and moved all table titles above their respective tables to follow standard formatting conventions.
[Updated in manuscript on page 7, Table 1 and page 10, table 2]

We have also added graphical representations as requested ( figure 1 page 2, figure 2 page 3, figure 3 page 4, figure 4 page 4, figure 5 page 5, and figure 6 page 9)

Reviewer 2 Report

Comments and Suggestions for Authors

The review article is good and structure very well. I would like to suggest only some minor suggestions like to include visual representation or any histological image or any clinical depiction of Endometriosis, moreover they can also give a graphical representation that represents the summary of this review where they can give an overview of pathways that are involved in effecting the infertility in context of Endometriosis.

Author Response

Comments: [Please include a visual representation—e.g., a histological or clinical image—of endometriosis. In addition, a graphical figure summarising the key pathways by which endometriosis affects infertility would greatly enhance the review.]

Response:
Thank you for this valuable suggestion. We have incorporated several visual representations  to address your request:

  1. Figure 1 (page 2): A graphical abstract summarising the principal pathways through which endometriosis impairs fertility. The schematic integrates chronic inflammation, oxidative stress, anatomical distortion, illustrating their interplay and downstream effects on oocyte quality, embryo development and implantation.
  2. Figure 2 (page 3): A high-resolution image of Peritoneal endometriosis as presented by Schippert et al. (2020) in their retrospective single-center analysis. Livid lesions and an increased vascular pattern are clearly visible, effectively highlighting its pathophysiology

  3. Figure 3 (page 4): type I and type II endemetriomas characterized by their cyst size 

  4. Figure 4 (page 4): a laparoscopic view of posterior compartment DIE 
  5. Figure 5 (page 5):  light microscopic view of morphological abnormalities such as a giant polar body that affect fertilization and early embryo progression 
  6. Figure 6 (page 9): A graphical representation of mechanical oocyte retrieval adopted from Baldini et al. (2019)

All these figures are now referenced in the text and have been uploaded in high-resolution TIFF format, meeting the journal’s specifications. We trust these additions will improve the manuscript’s clarity and visual impact.